# Astrocytes: Lessons Learned from the Cuprizone Model

**DOI:** 10.3390/ijms242216420

**Published:** 2023-11-16

**Authors:** Markus Kipp

**Affiliations:** Institute of Anatomy, Rostock University Medical Center, 18057 Rostock, Germany; markus.kipp@med.uni-rostock.de; Tel.: +49-(0)-381-494-8400

**Keywords:** myelin, astrocytes, glia, glue, multiple sclerosis, cuprizone, metabolics

## Abstract

A diverse array of neurological and psychiatric disorders, including multiple sclerosis, Alzheimer’s disease, and schizophrenia, exhibit distinct myelin abnormalities at both the molecular and histological levels. These aberrations are closely linked to dysfunction of oligodendrocytes and alterations in myelin structure, which may be pivotal factors contributing to the disconnection of brain regions and the resulting characteristic clinical impairments observed in these conditions. Astrocytes, which significantly outnumber neurons in the central nervous system by a five-to-one ratio, play indispensable roles in the development, maintenance, and overall well-being of neurons and oligodendrocytes. Consequently, they emerge as potential key players in the onset and progression of a myriad of neurological and psychiatric disorders. Furthermore, targeting astrocytes represents a promising avenue for therapeutic intervention in such disorders. To gain deeper insights into the functions of astrocytes in the context of myelin-related disorders, it is imperative to employ appropriate in vivo models that faithfully recapitulate specific aspects of complex human diseases in a reliable and reproducible manner. One such model is the cuprizone model, wherein metabolic dysfunction in oligodendrocytes initiates an early response involving microglia and astrocyte activation, culminating in multifocal demyelination. Remarkably, following the cessation of cuprizone intoxication, a spontaneous process of endogenous remyelination occurs. In this review article, we provide a historical overview of studies investigating the responses and putative functions of astrocytes in the cuprizone model. Following that, we list previously published works that illuminate various aspects of the biology and function of astrocytes in this multiple sclerosis model. Some of the studies are discussed in more detail in the context of astrocyte biology and pathology. Our objective is twofold: to provide an invaluable overview of this burgeoning field, and, more importantly, to inspire fellow researchers to embark on experimental investigations to elucidate the multifaceted functions of this pivotal glial cell subpopulation.

## 1. Introduction

The central nervous system (CNS) comprises neurons and neuroglial cells. Neurons receive and facilitate nerve impulses across their membranes to the next neuron, thus forming a fine-tuned communication network. Indispensable for fast and energy-efficient action potential propagation are the myelin sheaths built up by extended and modified plasma membranes of a specialized glia cell population—the oligodendrocytes [1]. Disorders of CNS myelin form a large and growing list of neurological disorders in humans, ranging from the most common myelin disease, multiple sclerosis (MS), to rare genetic conditions, such as Niemann–Pick disease. It is of note that oligodendrocyte dysfunctions also appear to be involved in different neurodegenerative and psychiatric disorders, including, among them, Alzheimer’s and Parkinson’s disease, eating disorders, depression, and schizophrenia [2,3,4,5,6].

MS is an autoimmune-mediated inflammatory disorder of the CNS with a still largely unknown aetiology. Two main clinical MS courses can be defined: a relapsing–remitting and a chronic, progressive course. Relapsing–remitting MS (RRMS) is characterized by recurrent episodes of clearly delineated clinical impairment (i.e., relapsing) from which the patients can eventually recover completely (i.e., remission). A relapsing–remitting disease course often turns into a secondary progressive (SPMS) course after approximately 10–15 years. In this secondary phase of the disease, individual relapses occur less frequently, but there is a slow, more or less continuous progression of the clinical impairment. The main histopathological features of MS are peripheral immune cell recruitment (id est, mostly monocytes and lymphocytes), blood–brain barrier integrity loss, reactive gliosis, oligodendrocyte damage, and, most importantly, demyelination. Mechanistically, it is believed that the disruption of the function of myelin results from an immunologically specific interaction between autoimmune T-/B-lymphocytes and myelin antigens [7]. 

Although demyelination is still a distinct pathological entity in MS, it has become increasingly apparent in the last two decades that substantial axonal and neuronal losses are equally important features. This phenomenon is at least partially related to the failure of remyelination. Remyelination is a regenerative process of the CNS. It occurs in three consecutive steps: (i) activation of oligodendrocyte progenitor cells (OPCs)—a widespread population of multipotent progenitors, (ii) their enrichment within areas of demyelination through migration and proliferation, and, finally, (iii) their differentiation into new myelinating oligodendrocytes [8]. It is unclear why, in some MS patients, remyelination is widespread, while, in others, it is sparse. Thus, understanding why a relatively robust regenerative process may lose *momentum* is an essential prerequisite for developing an effective therapeutic approach for this disease.

## 2. Glia Cells

Following the classical textbooks, neuronal cells can be divided into neurons and glial cells; the latter, based on their morphology, are subdivided into big (macroglia) and small (microglia) ones. While microglia constitute the innate immune cells of the CNS, macroglia are a diverse cell population, including oligodendrocytes, ependymal cells, pericytes, and astrocytes. 

Astrocytes are highly heterogeneous in form and function, including the protoplasmic astrocytes residing in the grey matter, fibrous astrocytes residing in the white matter, velate astrocytes, which are localized in brain regions where small neurons are densely packed (e.g., the olfactory bulb or the granular layer of the cerebellar cortex), radial glia cells with essential guidance functions for neurons during development, the cerebellar Bergmann glia [9,10], the retinal Müller glia [11,12], pituicytes that are localized in the neurohypophysis, Gomori-astrocytes, which are prominent in the arcuate nucleus of the hypothalamus [13], and perivascular astrocytes, whose endfeet connect with blood vessels and are fundamental for the establishment of the blood–brain barriers. Furthermore, the literature describes specialized astrocytes, which are observed only in the brain of higher primates, including interlaminar astrocytes [14] and varicose projection astrocytes [15]. 

Astrocytes assume various essential functions, much like assistants in a theatre play. For example, astrocytes structure neuronal networks and shield neurons from each other with their extensions. They, thus, function like set designers creating the backdrop of the stage. They are assistant directors, ensuring that the “actors” (the neurons) are where they need to be and interacting with the right colleagues at the right time [16,17]. Like oligodendrocytes [18], astrocytes supply neurons with metabolites and dispose of their waste products; thus, they are responsible for both the pantry and catering service, as well as sanitary facilities and waste disposal [19,20,21]. Furthermore, astrocytes maintain the finely tuned balance of ions and transmitters. Like props, they are responsible for ensuring that the performers have their lines and costumes ready and that unneeded materials are cleared away [22,23]. Finally, astrocytes receive signals from neurons and signal back to them, influencing synaptic transmission and being capable of modifying it. They are the prompters assisting the actors on the stage (adopted from *dasGehirn.info* (https://www.dasgehirn.info/grundlagen/glia/astrozyten-die-heimlichen-stars-des-gehirns?gclid=CjwKCAiA9dGqBhAqEiwAmRpTCzqdOdy2G26KS9fL-T4EQuAFkhV72cF7i1-9c0j4PRTz6yytgrLYYxoCy2kQAvD_BwE (accessed on 15 November 2023)) and [24]). 

Several excellent reviews have been written addressing the putative relevance of astrocytes in demyelinating disorders, including MS [25,26,27]. In this work, we will focus on astrocytic changes and their putative function in the so-called “cuprizone model,” which is frequently used to study mechanisms of oligodendrocyte degeneration, demyelination, and remyelination. Although the tabular listing of published works, which we provide at the end of this manuscript, claims to be comprehensive, we would like to focus in more detail on only a few works with groundbreaking insights for the field of research. The next chapter gives a brief overview of the most frequently used pre-clinical MS animal models, followed by a brief outline of the mechanistic and histo-pathological characteristics of the cuprizone model. 

## 3. Multiple Sclerosis Animal Models

To study the distinct aspects of MS pathology, several different animal models exist, and they are essential for assessing the impacts of novel therapeutic approaches. One category consists of the experimental autoimmune encephalomyelitis (EAE) models induced by injecting a CNS antigen combined with Freund’s and other adjuvants to trigger an immune response. From a motor–behavioral perspective, most of the EAE models lead to ascending paralysis in mice, with the severity and clinical course depending on factors like the immunization peptide and mouse strain used. If, for example, C56BL6 mice are immunized with the MOG_35–55_ (myelin oligodendrocyte glycoprotein) peptide, the animals develop a chronic disease course. In contrast, a relapsing–remitting disease course develops if SJL-mice are immunized with the PLP_139–151_ (proteolipid protein) peptide. Histopathologically, these models are characterized by multifocal inflammatory lesions, which can mainly be found in the spinal cord, brainstem, and cerebellum. 

Another group of commonly used animal models to understand MS disease pathology and to develop new therapeutic strategies are the toxin-mediated demyelination models. In these models, the toxins can be either administered via focal injection into the brain parenchyma (usually, lysophosphatidylcholine (LPC) is used) or provided to the animal per os (usually, cuprizone). LPC injection into the brain parenchyma causes rapid disruption of cellular membranes, myelin sheath disintegration, and focal demyelination within 1–2 weeks, followed by natural remyelination [28,29]. Reduced macrophage/microglial activation and myelin clearance post LPC injection in T-cell-deficient nude mice, along with inhibited remyelination in Rag-1-deficient mice lacking both B and T cells, indicate that the LPC model captures aspects of MS’s autoimmune component [30].

In contrast to the LPC model, demyelination in the cuprizone model occurs more slowly. Feeding young mice (around 8 weeks old) with cuprizone induces early apoptosis in oligodendrocytes, followed by microglia and macrophage activation and, finally, demyelination [31,32,33]. The exact mechanisms of oligodendrocyte death remain unclear, but it is believed to involve mitochondrial disruption due to a cuprizone-induced copper deficit, ferroptosis [34], and endoplasmic reticulum stress responses [35]. Emerging research indicates, however, that the detrimental effects of cuprizone may not be solely attributed to copper chelation or a selective impact on oligodendrocytes. Instead, it is proposed that a reactive cuprizone–copper complex is responsible for the toxic effects of cuprizone and that various cell types within the CNS are affected [36,37,38]. Why cuprizone predominately compromises oligodendrocytes is unknown, but the energy-intensive nature of myelin synthesis might play a role. Although the primary region of interest in many studies is the corpus callosum, demyelination in this model affects multiple white and grey matter brain areas [39]. During the 5–6 weeks of cuprizone exposure, the corpus callosum undergoes “acute demyelination.” When cuprizone intake is ceased at this time point, spontaneous remyelination occurs. If, however, the cuprizone intoxication is extended beyond 12–13 weeks, this leads to “chronic demyelination” with limited endogenous remyelination capacity [40]. All of the aforementioned animal models made, in the past, a significant contribution to the development of new therapeutic approaches in MS therapy. Of note is that some authors apply this model as well in the context of schizophrenia research, where myelin and oligodendrocyte abnormalities are well known [41,42]. 

In the next section, we will describe key papers investigating the response of astrocytes upon cuprizone-induced demyelination and functional studies that suggest that astrocytes can orchestrate de- and remyelinating pathways in this model. This section does not claim to discuss all publications relevant to astrocytes. Instead, we aim to provide a historical overview of how the function of astrocytes has been better understood over the past decades using the cuprizone model. In contrast, Table 1 lists all relevant works to provide the interested reader with an overview of the available literature.

## 4. Astrocytes and the Cuprizone Model

Astrocyte activation is a robust early event during cuprizone-induced demyelination [31]. Various laboratories have been able to demonstrate that during the course of cuprizone-induced demyelination, there is a significant upregulation in the expression of various astrocyte marker proteins, such as glial fibrillary acidic protein (GFAP), aldehyde dehydrogenase 1 family member L1 (ALDH1L1), and vimentin, within the demyelinated regions [43,44]. 

Figure 1 demonstrates findings published by Samuel Ludwin, one of the pioneers working with the cuprizone model from the late 1970s on. In Ludwin’s study, 3H-thymidine was used to label mitotic cells, revealing that alongside macrophages, astrocytes proliferate early during demyelination in the cerebellar peduncle, a frequently studied region in these initial periods of cuprizone research [45]. Despite the limitations of the applied method to label proliferating cells [46], these early studies demonstrated an early activation of astrocytes in the cuprizone model. Of note is that this first notion that prior to astrocyte activation, microglia cells respond to the cuprizone insult, was replicated by Matsushima’s lab in 1998 [47]. Some years later, when anti-GFAP labelling via immunohistochemistry became available, A. Mackenzie from the ARC Institute for Research on Animal Diseases in the UK demonstrated that astrocyte activation is widespread and not restricted to the cerebellar peduncles [39,48,49,50,51,52,53,54,55]. One of the first functional studies was published by Samuel Komoly and colleagues, who, in those days, was working at the National Institute of Neurological Disorders and Stroke, Bethesda/US, and, later, as the director of the neurology department at Pecs/Hungary. In that study, mice were intoxicated with cuprizone for 8 weeks to induce demyelination and, thereafter, provided normal chow to allow endogenous remyelination [56]. Throughout the periods of cuprizone treatment and recovery, brain tissue sections were subjected to hybridization and immunostaining with specific anti-Igf1 and Igf1-receptor probes to determine the precise locations and relative quantities of IGF-I and IGF-I receptor mRNAs and peptides. The authors found that in the white matter of untreated mice, there were no detectable levels of IGF-I or IGF-I receptor mRNAs or peptides. Conversely, in mice treated with cuprizone, astrocytes in regions where myelin was breaking down exhibited significantly elevated levels of both IGF-I mRNA and peptide. As the recovery process commenced, the expression of IGF-I in astrocytes decreased rapidly. At the same time, oligodendrocytes began to express the IGF-I receptor, suggesting that astrocyte-derived IGF1 regulates oligodendrocyte differentiation and remyelination. Although the results of subsequent studies suggested that IGF-1 is probably not a good candidate for treatment in MS [57,58], this was the first work in this model implicating that astrocytes orchestrate de- and remyelination in the diseased brain. In 2000, Brian Popkos’s lab showed that besides microglia accumulation, the extent of astrocyte activation might well be a good indicator of the overall extent of the cuprizone-induced pathology. In that study, the authors investigated the effect of IFN-gamma on myelin injury in the cuprizone model. They used a transgenic mouse line that expresses IFN-gamma under the transcriptional control of the *Mbp* gene. Careful evaluations revealed that mice overexpressing IFN-gamma preserved the myelin architecture, as demonstrated by LFB-PAS stains and anti-MBP immunohistochemistry. The assessment of glia reactivity showed that both microglia and astrocyte activation were less severe in the IFN-gamma-overexpressing mice, demonstrating that the quantification of glia reactivity is a good indicator of the overall extent of the cuprizone-induced pathology [59]. Our knowledge regarding possible cell–cell interactions in the cuprizone model involving astrocytes became more complex in the following years. In 2011, Matsushima’s lab showed that IL1ß-deficient mice have impaired remyelination capacities, which was paralleled by a lack of IGF-1 synthesis [60]. Because it has been shown in a previous study that astrocytes are the main source of IGF-1 in the cuprizone model, it was suggested that IL1ß might be necessary to induce IGF1 expression in astrocytes, which, in turn, promotes sufficient remyelination.

The cuprizone model also paved the way to imaging the development and progression of neuropathological processes via distinct imaging techniques, such as Positron emission tomography (PET). PET represents a sensitive method for visualizing neuropathological states, such as alterations in neuronal activity, deficiencies of neurotransmitter systems, or the extent of neuroinflammation. During PET, a small amount of a radioactive substance, known as a radiotracer, is introduced into the body. This radiotracer is usually a molecule chemically similar to a naturally occurring compound in the body, such as glucose or water, or it binds to endogenous proteins. As the radiotracer undergoes radioactive decay, it emits positrons, which can be visualized via gamma ray detection. Chen and colleagues were able to show that the mitochondrial translocator protein 18 kDa (TSPO), formerly known as the “peripheral benzodiazepine receptor,” is expressed in the cuprizone model by microglia and astrocytes [61]. Later studies, relying on this initial observation, used specific TSPO-radioligands, such as (18F)-GE180 [62] and others, to visualize activated microglia and astrocytes in living animals [43,63,64,65,66]. Zinnhardt and colleagues demonstrated that the expression of TSPO in either microglia or astrocytes is time-dependent in this model, with a predominant expression in microglia during demyelination and a shift towards astrocytic expression during remyelination [66]. Of note is that different TSPO radioligands, including (18F)-GE180, indicated equally good performance in MS patients [67,68]. Furthermore, the results of a recent study suggest that species-related differences should be considered when interpreting the results of such metabolic imaging techniques [69]. 

The importance of astrocytes for orchestrating de- and remyelinating events in the CNS was demonstrated in seminal work published by Stangels’s lab from Hannover/Germany. In that study, the authors used a mouse model in which thymidine kinase from the herpes simplex virus was targeted to astrocytes using the mouse GFAP promoter [70]. Ganciclovir functions as a nucleoside analogue, and it undergoes phosphorylation through the action of herpes simplex virus thymidine kinase within the GFAP-positive cell population. This phosphorylated form of ganciclovir competes with naturally occurring nucleotide triphosphates, thereby interfering with the process of DNA synthesis. Consequently, this disruption of DNA synthesis initiates apoptotic cell death, particularly in actively proliferating cells. Astrocyte ablation did not affect cuprizone-induced oligodendrocyte loss. However, without astrocytes, microglia recruitment and myelin debris clearance were hindered, leading to delayed remyelination.

In the last few years, single cell sequencing and/or single nuclei sequencing techniques were introduced, and they now allow the scientific community to investigate translatome changes of various cell types in the cuprizone model, including astrocytes. Hou and colleagues performed single-nucleus RNA sequencing from three distinct experimental categories: (1) a demyelination group, which underwent a 5-week cuprizone treatment, (2) a remyelination group, exposed to a 5-week cuprizone treatment followed by a 2-week regular chow diet, and (3) the control group, which received regular chow throughout the entire experimental period. As expected, the relative frequency of mature oligodendrocytes decreased during demyelination and recovered during remyelination [71]. Surprisingly, the frequency of astrocyte nuclei slightly dipped during demyelination but significantly recovered during remyelination. On the first view, this contrasts with the histological finding that severe astrocytosis can be observed after acute cuprizone-induced demyelination. Of note is that in this study, nuclei isolated from the corpus callosum and cortex were analyzed. Under control conditions, most cortical astrocytes express very low, almost undetectable GFAP levels, with a robust increase during demyelination. As such, the robust increase in GFAP^+^ cell numbers during cuprizone-induced demyelination does not really reflect the accumulation of new astrocytes. Indeed, it has been shown that pro-apoptotic pathways are activated in astrocytes after acute cuprizone-induced demyelination [35], which would explain the observed dip in the frequency of astrocyte nuclei. On the other hand, astrocyte proliferation has been demonstrated in the cuprizone model by several labs [45,72,73], so there remains a certain uncertainty about how to interpret these data. Nevertheless, although astrocyte abundance was minimally affected by cuprizone intoxication, marked transcriptional changes were evident during de- and remyelination. The most important findings of this single-nucleus RNA sequencing study can be summarized as follows. Firstly, homeostatic astrocytes are diverse at a steady state, with subpopulations expressing genes related to the extracellular matrix or neurogenesis. Nine subclusters were identified. Secondly, during demyelination, each subpopulation uniquely upregulated stress responses and mTOR pathway genes. However, certain pathways, such as the IFN-R pathway, were upregulated only in specific subpopulations. Thirdly, in the remyelination phase, pro-inflammatory clusters decreased, while some homeostatic astrocyte clusters were replaced by others. Of note is that the study also provided evidence that only minor expressional changes occur in neurons, verifying our own results that at least after acute cuprizone-induced demyelination, the degeneration of entire neurons, despite the high frequency of acute axonal injury [74], is not a characteristic feature in this model [75].

Schröder et al. applied the ribosomal tagging (RiboTag) approach to obtain insight into astrocyte function in the cuprizone model [76]. The RiboTag approach is a molecular technique used to study gene expression and protein translation within specific cell populations in complex tissues. First, transgenic animals are generated—usually mice—in which a specific ribosomal protein (often Rpl22) is genetically engineered to be tagged with a sequence (in this case, hemagglutinin A or HA), which allows for efficient isolation and purification of ribosome-bound mRNA and associated translating ribosomes. This tagged ribosomal protein is expressed under the control of a cell-type-specific promoter (in that study, under the GFAP promotor), so it is only present in a specific subset of cells within the animal. The transgenic animals are then bred or crossed with other genetically modified mice that express Cre recombinase under the control of a cell-type-specific promoter. This results in the deletion of a “floxed-stop” cassette that was preventing the expression of the tagged ribosomal protein. As a result, the tagged ribosomal protein is now expressed in the desired cell type. Tissues or cells of interest can now be sampled from these transgenic animals and subsequently analyzed. 

Schröder and colleagues performed HA-tagged ribosome isolation from four distinct experimental categories: a demyelination group, which underwent a 5-week cuprizone treatment; two remyelination groups, which were exposed to 5-week cuprizone treatment followed by a 0.5- and 2-week regular chow diet; and a control group, which received regular chow throughout the entire experimental period. Upon isolation of the corpus callosum, HA-tagged ribosomes were isolated from the tissue lysate, and RNA was isolated from the HA-tagged ribosomes. Of note is that under conditions of acute demyelination (id est, 5-week cuprizone intoxication), basically all HA^+^ cells also expressed GFAP, which means that under such conditions, RiboTag mice are suitable for revealing astrocyte-specific transcriptomic signatures. Differential expression analysis of samples from 5-week-cuprizone-fed mice versus controls revealed 1453 differentially expressed genes, of which 1041 were up- and 412 down-regulated. Within the group of differentially upregulated genes, there were numerous chemokines (for example, Ccl6, Cxcl5, and Cxcl10), genes encoding for markers of cell activation (Cd86), and interleukins (Il3ra). Of note is that the results of recent studies suggest that CXCL10 orchestrates the activation of microglia in the cuprizone model [73,77]. In line with the single-nucleus RNA sequencing results from Hou and colleagues, in depth analysis of the RiboTag experiments suggests that astrocytes actively contribute to inflammation during demyelination but adopt a regenerative phenotype during remyelination. Another important finding from Schröder and colleagues was the existence of distinct expression signatures of HA-tagged cells during early and late remyelination, suggesting that astrocytes shape their adoptive phenotype according to the dynamic changes of oligodendrocytes during maturation and remyelination [76]. As a limitation of that study, under control conditions, approximately half of the HA^+^ cells were positive for APC/CC1 (a frequently used maker of mature oligodendrocytes) but did not express GFAP. The authors suggest that these HA^+^APC^+^GFAP^-^ cells are mature oligodendrocytes that did express GFAP during their development [78,79]. 

## 5. Conclusions

For a long time, astrocytes lived in the shadows in neuroscience. After an initial focus on neurons, research turned its attention to the myelin producers of the central nervous system, the oligodendrocytes, and the cells of innate immunity, the microglia. However, astrocytes are gradually stepping out of the shadows. It is becoming increasingly clear that astrocytes do not represent a uniform cell population but are highly heterogeneous, with various essential functions. Without astrocytes, normal development and maintenance of neuronal function would not be possible. They are not just a filler substance holding the nervous system together, but rather, like ministers in a government, working diligently behind the scenes to enable higher cognitive functions. Furthermore, astrocytes play an important role in the development and progression of a number of neurological diseases, and we are just at the beginning of understanding the relevance and function of astrocytes in such diseases. This is, maybe, best illustrated with neuromyelitis optica, a neurological disorder that was, for a long time, considered to be a subvariant of MS. Today, we know that an autoimmune attack against the water channel protein aquaporin 4 is a key event during neuromyelitis optica, and this protein is expressed by astrocytes [80,81].

What did we learn by using the cuprizone model? In fact, various studies using this model were able to show that astrocytes are functionally involved in the process of oligodendrocyte damage and demyelination. Proteins expressed by astrocytes and functionally relevant for the cuprizone-induced pathological changes are, among others, the Transient Receptor Potential Ankyrin 1 (TRPA1), a nonselective cation channel with relatively high Ca^2+^ permeability [82], Lipocalin 2 (LCN2), also known as oncogene 24p3 or neutrophil gelatinase-associated lipocalin (NGAL), an adipocytokine implicated in various immunological functions [83], lymphotoxin-alpha, a cytotoxic protein [84], and the IκB kinase 2 pathway, which induces nuclear factor kappa B activation [85]. Regarding a possible function of astrocyte-expressed factors in the context of remyelination, among others, galectin 3 [86], the CXCR4 ligand, CXCL12 [87] BDNF [88], and the Cav1.2 voltage-gated Ca^2+^ channel [88] have been identified.

Whether astrocytes are beneficial or detrimental in the context of MS is discussed controversially, and this has been elaborated in several excellent review articles in the past [89,90]. On the one hand, available data suggest that astrocytes help protect demyelinated tissue from additional harm and mitigate the negative impact of neuroinflammatory activities on these tissues through the release of anti-inflammatory cytokines, like TGF-β and interleukins 10 and 27. These actions include blocking inflammatory cells from entering demyelination zones and fostering an environment conducive to remyelination. On the contrary, some studies challenge this view by attributing to astrocytes a role in exacerbating demyelination due to their release of chemokines that draw inflammatory microglial cells to the sites of injury, thereby hindering the remyelination process. Results in the cuprizone model are equally inconclusive. Two studies addressed this question by ablating astrocytes and studying the consequences for myelin repair. Skripuletz and colleagues used Ganciclovir treatment in GFAP–thymidine kinase transgenic mice to ablate astrocytes [73]. While astrocyte ablation did not influence the cuprizone-induced loss of oligodendrocytes, the removal of damaged myelin sheaths by microglia appears to be impaired, thus inhibiting the regeneration of oligodendrocytes and myelin. While Skripuletz and colleagues ablated astrocytes during active demyelination, Madadi and colleagues ablated astrocytes with La-aminoadipate (L-AAA) after chronic cuprizone-induced demyelination [91]. Under such experimental conditions, ablated animals showed better myelin and functional recovery, suggesting that astrocytes impair myelin repair in the more chronic lesion. The results of both studies are difficult to compare. On the one hand, different experimental approaches were used to ablate astrocytes (L-AAA versus thymidine kinase). On the other hand, the ablation of astrocytes was carried out at two entirely different points in time. In the first study, astrocytes were removed from a highly inflammatory environment characterized by a large amount of myelin debris, significant activation of microglia, and infiltration by oligodendrocyte precursor cells. In the second study, in contrast, astrocytes were removed from a less inflammatory environment containing no or only limited myelin debris, less pronounced microglia activation, and sparsely oligodendrocyte precursor cells. Based on our current knowledge, it appears that astrocytes initially play a supportive role in myelin regeneration, which, however, can turn negative as the lesion becomes chronic. Future studies will show whether such a simplified statement is indeed true. 

**Table 1 ijms-24-16420-t001:** IHC (Immunohistochemistry); ISH (in situ hybridization); IGF-1 (Insulin-like growth factor 1); IFN-γ (Interferon-gamma); MIP-1alpha (Macrophage Inflammatory Protein-1 Alpha); IL-1β (Interleukin-1β); MHC (Major Histocompatibility Complex); PDGF-A (Platelet-Derived Growth Factor A); OPC (Oligodendrocyte Progenitor Cells); Notch-1 (Neurogenic locus notch homolog protein 1); ADAM12 (ADAM Metallopeptidase Domain 12); CXCL12 (C-X-C motif chemokine 12); Cx47 (Connexin 47); APP (Amyloid precursor protein); TSPO (Translocator protein); COX-1 (Cyclooxygenase-1); FABP7 (Fatty Acid Binding Protein 7); MMP-3 (Metalloproteinase-3); RXRβ (Retinoid X Receptor β); Tumor necrosis factor receptor 1 (TNFR1); GFAP (Glial fibrillary acidic protein); IL6 (Interleukine 6); mGluR (metabotropic glutamate receptor); BDNF (brain derived neurotrophic factor); OSMR (Oncostatin M receptor); NRF2 (Nuclear factor erythroid-2-related factor 2); S1P (Sphingosine-1-Phosphate); NF-κB (nuclear factor k-light-chain-enhancer of activated B cells); GLI1 (GLI Family Zinc Finger 1); SOX10 (SRY-box transcription factor 10); DDIT3 (DNA Damage Inducible Transcript 3); HO-1 (Hämoxygenase-1); PI3K (Phosphoinositide 3-kinase); CTNF (Ciliary neurotrophic factor); PAR-1 (Protease-activated receptor-1); GDNF (Glial cell line-derived neurotrophic factor); P2X7 (purinoceptor 7); 8-OHdG (8-hydroxy-2’ –deoxyguanosine); SIRT-1 (NAD-dependent protein deacetylase sirtuin-1); AQP4 (Aquaporin-4); NgR1 (Nogo-66 receptor 1); SPARC (Secreted *protein* acidic and rich in cysteine); EBI-2 (EBV-induced gene 2); TrkB (Tropomyosin receptor kinase B); CTR1 (High affinity copper uptake protein 1); ATP7A (ATPase copper transporting alpha); FTH1 (Ferritin Heavy Polypeptide 1); GLAST1 (GLutamate ASpartate Transporter 1); LCN2 (Lipocalin-2); ALDH1L1 (aldehyde dehydrogenase 1 family, member L1); TIMP1 (TIMP metallopeptidase inhibitor 1); TRAP1 (TNF Receptor Associated Protein 1); PTPRZ (Protein Tyrosine Phosphatase Receptor Type Z1); HNK-1 (Human natural killer-1).

Citation	Main Finding(s)
[46]	Astrocyte proliferation via 3H-thymidine labelling
[48]	Astrocyte activation, demonstrated by IHC and ISH
[56]	IGF-1 is expressed by astrocytes and the receptor by oligodendrocytes
[92]	Altered astrocytic glutathione-S-transferase isoform expression during demyelination
[93]	Altered astrocytic glutathione-S-transferase isoform expression during remyelination
[47]	Astrogliosis promptly follows microgliosis during demyelination
[59]	Amelioration of cuprizone-induced pathology ameliorates the extent of astrocyte activation; IFN-γ overexpression, driven by the MBP reporter
[94]	Amelioration of cuprizone-induced pathology ameliorates the extent of astrocyte activation; MIP-1alpha deficiency
[60]	IL1β-deficient mice have lower IGF-1 levels during the remyelination phase
[95]	Astrocytes express MHC class I and II
[96]	Transgenic mice that overexpress PDGF-A in astrocytes have increased OPC numbers
[61]	Peripheral benzodiazepine receptor is expressed by astrocytes and microglia
[97]	Osteopontin is expressed by astrocytes and microglia
[98]	Notch1 is expressed by various cell types, including astrocytes, within remyelinating lesions
[84]	Lymphotoxin-alpha is expressed by astrocytes and exacerbates demyelination
[99]	Metallothionein-I and –II are expressed by astrocytes
[100]	Different pathologies, including axonal injury and astrocyte activation, are more pronounced in aged versus young mice during demyelination
[101]	Complement regulatory protein Crry overexpression in astrocytes protects against demyelination
[102]	The acyl-CoA synthetase, lipidosin, is expressed by astrocytes
[40]	Platelet-derived growth factor-A overexpression in astrocytes supports remyelination
[103]	Metallothionein I/II are expressed by astrocytes
[39]	Extent of astrocytosis differs between demyelinated white and grey matter areas
[104]	Cortical demyelination, but not astrogliosis per se, is associated with accelerated cortical spreading depression
[105]	Astrocyte progenitor cells accumulate in cuprizone lesions
[106]	ADAM12 is expressed by astrocytes
[107]	Increased numbers of astrocytes in vivo within the subventricular zone during demyelination, and numbers were decreased by intraventricular Noggin infusion
[108]	CXCL12 is expressed by astrocytes and microglia
[109]	Cx47 is expressed by astrocytes
[110]	Glial isoform of APP is expressed by astrocytes
[111]	C3a and C5a overexpression exacerbates demyelination and delays remyelination
[112]	IV-injected human-embryonic-stem-cell-derived neural precursor cells into mice express GFAP to a limited extent
[85]	IκB kinase 2 depletion in astrocytes ameliorates demyelination
[113]	Smad1, Smad5, and Smad8, intracellular effectors of the bone morphogenetic protein (BMP) family of proteins, are active in oligodendrocytes and a subset of astrocytes
[65]	TSPO is expressed by astrocytes and microglia
[114]	COX-1 is expressed by astrocytes and microglia
[86,115]	Galectin-1 and -3 are expressed by astrocytes and microglia; galectin-3-deficient mice show impaired remyelination
[116]	Serine palmitoyltransferase, the rate-limiting enzyme for ceramide de novo biosynthesis, is expressed by astrocytes
[117]	FABP7 is expressed by astrocytes
[118]	IGF1 is expressed by astrocytes
[119]	MMP3 and MMP9 are expressed by astrocytes
[52]	RXRβ is expressed by astrocytes
[120]	Carbonic Anhydrase II is expressed by astrocytes
[121]	Act1-deletion in astrocytes ameliorates demyelination
[122]	p65 is active in astrocytes
[87]	TNFR1 and TNFR2 are expressed by astrocytes and microglia; CXCL12 is expressed by astrocytes, which promotes OPC proliferation and differentiation
[72]	In contrast to microgliosis, astrocytosis persists during de- and remyelination. Astrocyte reaction is characterized, among others features, by early astrocyte proliferation and increased expression of GFAP, vimentin, and fibronectin. Furthermore, there is an elaboration of a dense network of processes
[73]	Astrocyte ablation results in impaired remyelination
[123]	IGF1 infusions can decrease astrocyte numbers during remyelination
[124]	Glutamate-aspartate transporter is expressed by astrocytes
[125]	Receptor protein tyrosine phosphatase β is expressed in astrocytes
[126]	IL6 is expressed by astrocytes
[88]	mGluR1, mGluR5, and BDNF are expressed by astrocytes; astrocyte-derived BDNF promotes recovery from cuprizone-induced demyelination
[77]	CXCL10 is mainly expressed by astrocytes
[127]	Oncostatin M receptor is expressed by astrocytes and microglia; OSMR deficiency aggravates demyelination; CNS-targeted OSM treatment ameliorates demyelination
[128]	Erk is especially activated in astrocytes and promotes demyelination
[129]	Transgenic mice that overexpress IL6 in astrocytes show reduced glia activation, axonal injury, and OPC recruitment
[82]	Transient Receptor Potential Ankyrin 1 (TRPA1) is expressed in astrocytes; TRPA1 deficiency significantly attenuated cuprizone-induced demyelination by reducing the apoptosis of mature oligodendrocytes
[130]	Activation of the astrocytic Nrf2/ARE system ameliorates demyelination
[131]	Transgenic mice that overexpress IL-17A in astrocytes show aggravation of demyelination
[132]	Transglutaminase 2 is expressed by astrocytes
[133]	Transgenic mice that overexpress IL6 in astrocytes show amelioration in the cuprizone-induced pathologies
[134]	S1P receptor 1 is expressed by astrocytes, and its modulation ameliorates demyelination
[135]	Astrocytes show NF-κB activation
[136]	Gli1 is expressed by astrocytes after chronic, but not acute, demyelination
[137]	Cuprizone induces astrocyte atrophy in the rat
[138]	Metallothionein I/II and Megalin are expressed by astrocytes
[139]	Sox10 converts astrocytes into oligodendrocyte-like cells
[140]	Transplanted astrocytes convert into oligodendrocyte-like cells
[35]	DDIT3 is expressed by oligodendrocytes and astrocytes
[141]	Sox2 converts astrocytes into oligodendrocyte-like cells
[91]	Astrocyte ablation augments remyelination after chronic demyelination
[63]	TSPO is expressed by microglia and astrocytes
[142]	Transferrin can be incorporated by all glial cells among astrocytes
[143]	Overexpression of GFAP reduces cuprizone-induced apoptosis, demyelination, and acute axonal damage
[144]	Mesenchymal stem cells reduce astogliosis and microgliosis
[145]	CD38 is expressed by astrocytes and microglia; CD38 ameliorates demyelination
[146]	Astrocytes express NRF2, HO-1, and PI3K; Ginkgolide K augments the expression of these proteins
[147]	Astrocytes express SOX2, CNTF, IGF2, and BDNF
[148]	Astrocyte-specific deletion of Transient receptor potential ankyrin 1 delays demyelination
[149]	PAR1 knock-out mice demonstrate skewing of reactive astrocyte signatures towards a pro-repair phenotype
[150]	Astrocytes express BDNF and GDNF; Ginkgolide B augments the expression of these proteins
[151]	Deletion of astrocytic Cav1.2 channels leads to reduced astrocytosis, microgliosis, ameliorates inflammation, and promotes remyelination
[152]	P2x7 receptors are expressed by astrocytes and microglia; demyelination is ameliorated in P2x7-deficient mice
[153]	Induced neural stem cells ameliorate astrocytosis
[154]	CD44 is expressed by astrocytes and microglia; Cd44 deficiency does not ameliorate cuprizone-induced pathology
[155]	Some astrocytes show oxidative damage to DNA (id est, 8-OHdG^+^)
[156]	SIRT1 is expressed by astrocytes, microglia, and mature oligodendrocytes
[157]	Mesenchymal stem cell transplantation ameliorates astrocytosis
[158]	AQP4 is expressed in astrocyte endfeet; polarized expression is reduced after demyelination
[159]	Astrocytes phagocytose myelin; astrocytes express BDNF, CTNF, Nestin, SOX2, Notch, and ß-catenin; expression profiles are regulated by ethyl pyruvate
[160]	Astrocytes express NgR1, SPARC, and Hevin; astrocytic NgR1 sublocalization alters during demyelination;
[161]	Astrocytes express C3; TIC knock-outs show reduced C3 expression
[162]	Aastrocyte participation in the tripartite synapse during demyelination
[163]	mGluR5 is expressed by astrocytes and orchestrates remyelination
[164]	EBI2 receptor is expressed in astrocytes and microglia
[165]	Astrocytes express TrkB, CTR1, ATP7A, and ATP7B; demyelination is ameliorated in mice lacking astrocytic TrkB expression
[166]	Fth deletion in Glast1/EAAT1-positive astrocytes inhibits remyelination
[167]	EAAT2 is expressed by astrocytes
[168]	Astrocytes express S100B, GFAP, vimentin, LCN2, and ALDH1L1
[83]	LCN2 is expressed by astrocytes; oligodendrocyte loss is more severe in Lcn2^-/-^ animals
[169]	Ongoing astrocytosis weeks after completion of remyelination
[170]	C3d, S100a10, Stat3, and Timp1 are expressed by astrocytes
[171]	C3d and S100a10 are expressed by astrocytes; Bu Shen Yi Sui capsules promote an A2 phenotype
[172]	Combined mesenchymal stem cell transplantation and astrocyte ablation support remyelination
[173]	Cuprizone intoxication induces astrocyte endfeet sweelings
[174]	Astrocytes express TRAP1; β-hydroxybutyrate downregulates the expression of this protein
[71]	High-resolution, single-nucleus RNA sequencing (snRNA-seq) analysis of gene expression changes across various brain cells, including astrocytes
[175]	Bone Marrow Mesenchymal Stem Cells reduce astrocytosis
[176]	Astrocytes express HNK-1-O-Man+ PTPRZ
[177]	A1 versus A2 astrocyte expression profiling
[178]	The AQP4 inhibitor TGN020 ameliorates astrocyte and microglia activation
[76]	Astrocytic transcriptome signature investigated via ribosomal tagging

## Figures and Tables

**Figure 1 ijms-24-16420-f001:**
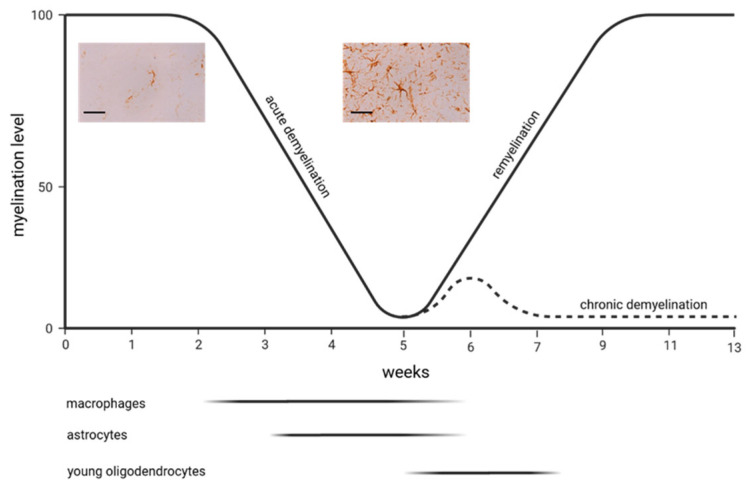
The proliferation of different glia subpopulations during the course of cuprizone-induced demyelination [45]. The curved line in the graph illustrates the myelination levels of the midline of the corpus callosum during acute demyelination (week 5) and the subsequent remyelination phase as the animals are provided normal chow after acute demyelination. The dotted line illustrates the myelination levels of the midline of the corpus callosum if the intoxication with cuprizone is continued until week 13 (chronic demyelination). The straight lines beneath the graph illustrate the periods of glia cell proliferation. The inserts show cortical astrocytes, visualized using anti-GFAP immunohistochemistry, in control (left image) and 5-week-cuprizone-intoxicated mice (right image). The inserted bars are 50 µm.

## Data Availability

Not applicable.

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
