# Peer review of "Astrocytes: Lessons Learned from the Cuprizone Model"

_ijms, 2023, doi:10.3390/ijms242216420_

Round 1
Reviewer 1 Report
Comments and Suggestions for Authors
The authors propose a review, as explained in the title, which aims to describe how the research carried out with the cuprizone animal model has so far clarified the role of astrocytes in demyelination and remyelination in neurological pathologies. This purpose is clearly reported in the abstract and in the first paragraphs of the paper.
“…we systematically synthesize the findings from various cuprizone studies that delve into the cellular responses and modulatory roles of astrocytes in the context of myelin degeneration and repair”
“In this work, we will focus on astrocytic changes and their putative function in the so called “cuprizone-model” which is frequently used to study mechanisms of oligodendrocyte degeneration, demyelination and remyelination”
“we will describe key papers investigating the response of astrocytes upon cuprizone-induced demyelination and functional studies which suggest that astrocytes can orchestrate de- and remyelinating pathways in this model.”
However, as indicated at the end of paragraph 3, the paper provides a historical vision of some different methodologies used to study astrocytes in the cuprizone model. A few works are then discussed in detail, which mostly focus on the proliferation and generic activation of the astrocyte during the experimental model. Therefore, from what is suggested in the first paragraphs, we expect a discussion of the most relevant in vivo studies that demonstrate effective mechanisms activated by/in astrocytes during the pathological process. In this sense, the content of the work does not reflect what proposed as the purpose of the review.
-At page 3, lines 131-133, author claims that “The exact mechanisms of oligodendrocyte death remain unclear, but it's believed to involve mitochondrial disruption due to a cuprizone-induced copper deficit, ferroptosis [29] and endoplasmatic reticulum stress responses [30]”.
The mechanism of toxicity of cuprizone is still not completely understood, and several studies suggest that cuprizone's toxicity is not due to copper depletion or specific targeting of oligodendrocytes, but rather to an active CPZ-copper complex, which also acts on other cell types of the CNS (Pasquini et al, Neurochem.Res 2007; Yamamoto et al, J. Mol. Struct. Theochem 2009; Taraboletti et al., Biochemistry 2017. Morgan et al, ASN Neuro 2022), with a mechanism that has yet to be clarified. The authors must describe the experimental model taking into account this evidence, as it can have a profoundly impact on the role of astrocytes in demyelination processes.
-The final paragraph is inconclusive, and shifts the focus, also without providing any particular detail, on the involvement of AQP4 in neuromyelitis optica.
Author Response
Reviewer #1:
Question 1: Thank you very much for taking the time to review this work. I hope that I was able to implement your valuable suggestions, thereby improving the quality of the manuscript. You correctly stated that the authors propose a review, as explained in the title, which aims to describe how the research carried out with the cuprizone animal model has so far clarified the role of astrocytes in demyelination and remyelination in neurological pathologies. This purpose is clearly reported in the abstract and in the first paragraphs of the paper. However, as indicated at the end of paragraph 3, the paper provides a historical vision of some different methodologies used to study astrocytes in the cuprizone model. A few works are then discussed in detail, which mostly focus on the proliferation and generic activation of the astrocyte during the experimental model. Therefore, from what is suggested in the first paragraphs, we expect a discussion of the most relevant in vivo studies that demonstrate effective mechanisms activated by/in astrocytes during the pathological process. In this sense, the content of the work does not reflect what proposed as the purpose of the review.
Answer 1: Thank you very much for this valuable comment. Indeed, we recognize that what is mentioned in the abstract and the initial parts of this review article only partially reflect the subsequent content. We have adopted the sections not provoking misleading expectations. Beyond that, we have significantly adopted the conclusion sections to meet your expectations.
Question 2: -At page 3, lines 131-133, author claims that “The exact mechanisms of oligodendrocyte death remain unclear, but it's believed to involve mitochondrial disruption due to a cuprizone-induced copper deficit, ferroptosis [29] and endoplasmatic reticulum stress responses [30]”. The mechanism of toxicity of cuprizone is still not completely understood, and several studies suggest that cuprizone's toxicity is not due to copper depletion or specific targeting of oligodendrocytes, but rather to an active CPZ-copper complex, which also acts on other cell types of the CNS (Pasquini et al, Neurochem.Res 2007; Yamamoto et al, J. Mol. Struct. Theochem 2009; Taraboletti et al., Biochemistry 2017. Morgan et al, ASN Neuro 2022), with a mechanism that has yet to be clarified. The authors must describe the experimental model taking into account this evidence, as it can have a profoundly impact on the role of astrocytes in demyelination processes.
Answer 2: Thank you for this fruitful comment. We have adopted section 3 of the manuscript following your suggestions.
Question 3: The final paragraph is inconclusive and shifts the focus, also without providing any particular detail, on the involvement of AQP4 in neuromyelitis optica.
Answer 3: As already pointed out above, we have substantially modified the last section of the manuscript. Nevertheless, the discovery that NMO is due to a direct autoimmune attack against an astrocytic protein is relevant to the central message of the article and thus should be mentioned. We hope that the kind reviewer can follow our argumentation. Nevertheless, we have included a citation referencing this topic.
Reviewer 2 Report
Comments and Suggestions for Authors
The review article submitted focus on changes and putative function of astrocytes in cuprizone-model which a well establish experimental setup to study mechanisms of oligodendrocyte degeneration, demyelination remyelination. On this topic I regommend to cite: “Blakemore, W.F. Observations on oligodendrocyte degeneration, the resolution of status spongiosus and remyelination in cuprizone intoxication in mice. J Neurocytol 1, 413–426 (1972). https://doi.org/10.1007/BF01102943” – as far as I know Blakemore was the first who described histo-102 pathological characteristics of the cuprizone model.
Author Response
Reviewer #2:
Question 1: The review article submitted focus on changes and putative function of astrocytes in cuprizone-model which a well establish experimental setup to study mechanisms of oligodendrocyte degeneration, demyelination remyelination. On this topic I recommend to cite: "Blakemore, W.F. Observations on oligodendrocyte degeneration, the resolution of status spongiosus and remyelination in cuprizone intoxication in mice. J Neurocytol 1, 413–426 (1972). https://doi.org/10.1007/BF01102943 " [Titel anhand dieser DOI in Citavi-Projekt übernehmen] – as far as I know Blakemore was the first who described histopathological characteristics of the cuprizone model.
Answer 1: Thank you very much for taking the time to review this work. Thank you for the comment. We have adopted the citations accordingly.
Reviewer 3 Report
Comments and Suggestions for Authors
The author summarized the latest finding on the role of astrocyte in cuprizone induced demyelination model. This paper presents a balanced update of the findings obtained by many researchers, which is well-written review in the topic.
This reviewer has some questions for the author, a leading authority in this field.
1 The author descripted that oligodendrocyte dysfunctions is involved in various neurodegenerative and psychiatric disorders. Some studies used cuprizone model as a Schizophrenia. Are these appropriate ? It is not necessary to add these sentences to the manuscript.
2 The author written that ‘’Astrocytes are highly heterogeneous in form and function.’’ But there are no description about Bergmann glia and the retinal Müller glia in Reference 9 – 11. This reviewer recommend adding appropriate references.
3 It is known that astrocyte can differentiate into proinflammatory A1 and protective A2 types. The depletion of A2 astrocyte would aggravate oligodendrocyte loss. However, why astrocyte ablation did not affect cuprizone-induced oligodendrocyte loss ?
Author Response
Reviewer #3:
Question 1: Thank you very much for taking the time to review this work. The author descripted that oligodendrocyte dysfunctions is involved in various neurodegenerative and psychiatric disorders. Some studies used cuprizone model as a Schizophrenia. Are these appropriate ? It is not necessary to add these sentences to the manuscript.
Answer 1: Indeed, several authors use this model to study schizophrenia-related pathological changes. In my opinion, there is convincing evidence that myelin and oligodendrocyte abnormalities are an essential aspect of this and other psychiatric disorders. One perplexing aspect of schizophrenia is the extended delay, sometimes up to two decades, between the initial neuropathological changes and the onset of clinical symptoms. This delay could be due to abnormalities in the function of oligodendrocytes, which are responsible for myelination. If myelination is defective from the beginning, it could lead to synaptic dysfunction and aberrant neural pathways that are fundamental to schizophrenia. It is theorized that the latency might stem from events happening as early as the prenatal stage or during infancy when abnormal oligodendrocytes that myelinate might first emerge. Since the population of these cells can double approximately every two years, there could be a thousand-fold increase by the time an individual reaches twenty, which is often when the clinical symptoms become apparent. In such cases, the protracted latency period results from the gradual increase in the volume of abnormally myelinated neural pathways, which, after a considerable time, reaches a threshold that manifests the complete clinical picture of schizophrenia. Medications like fingolimod or siponimod, which aid oligodendrocytes and neural pathways, could prove beneficial. Following your comment, we have adopted the manuscript now mentioning that some authors apply this model as well in the context of schizophrenia research, where myelin and oligodendrocyte abnormalities are well known.
Question 2: The author written that ‘’Astrocytes are highly heterogeneous in form and function.’’ But there are no description about Bergmann glia and the retinal Müller glia in Reference 9 – 11. This reviewer recommend adding appropriate references.
Answer 2: Thank you for this fruitful comment. We have modified the manuscript accordingly and added appropriate citations for both cell populations.
Question 3: It is known that astrocyte can differentiate into proinflammatory A1 and protective A2 types. The depletion of A2 astrocyte would aggravate oligodendrocyte loss. However, why astrocyte ablation did not affect cuprizone-induced oligodendrocyte loss?
Answer 3: Thank you for this fruitful comment. The A1/A2 concept does not fit into MS biology, where, during remyelination, a certain level of inflammation is pivotal, allowing the removal of myelin debris. The cuprizone-induced oligodendrocyte loss is a cell-intrinsic driven process, where astrocytes and microglia play little roles. In contrast, the subsequent demyelination and axonal injury processes may well be shaped by the astrocytes and microglia. Indeed, Skripuletz and colleagues demonstrated this in their BRAIN paper. We have added a summary of these findings in the last chapter of the manuscript.
Round 2
Reviewer 1 Report
Comments and Suggestions for Authors The authors have adequately replied to the issues and deepened the text accordingly.